# Resonance-enhanced three-photon luminesce via lead halide perovskite metasurfaces for optical encoding

Yubin Fan [1,4], Yuhan Wang[1,4], Nan Zhang[1], Wenzhao Sun[1], Yisheng Gao[1], Cheng-Wei Qiu [2], Qinghai Song [1,3] & Shumin Xiao[1,3]

Lead halide perovskites have emerged as promising materials for photovoltaic and optoelectronic devices. However, their exceptional nonlinear properties have not been fully exploited in nanophotonics yet. Herein we fabricate methyl ammonium lead tri-bromide perovskite metasurfaces and explore their internal nonlinear processes. While both of third-order harmonic generation and three-photon luminescence are generated, the latter one is less affected by the material loss and has been significantly enhanced by a factor of 60. The corresponding simulation reveals that the improvement is caused by the resonant enhancement of incident laser. Interestingly, such kind of resonance-enhanced three-photon luminescence holds true for metasurfaces with a small period number of 4, enabling promising applications of perovskite metasurface in high-resolution nonlinear color nanoprinting and optical encoding. The encoded information 'NANO' is visible only when the incident laser is on-resonance. The off-resonance pumping and the single-photon excitation just produce a uniform dark or photoluminescence background.

[1] State Key Laboratory on Tunable laser Technology, Ministry of Industry and Information Technology Key Lab of Micro-Nano Optoelectronic Information System, Shenzhen Graduate School, Harbin Institute of Technology, Shenzhen 518055, China. [2] Department of Electrical and Computer Engineering, National University of Singapore, 4 Engineering Drive 3, Singapore 117583, Singapore. [3] Collaborative Innovation Center of Extreme Optics, Shanxi University, Taiyuan 030006, China. [4] These authors contributed equally: Yubin Fan, Yuhan Wang. Correspondence and requests for materials should be addressed to C.-W.Q. (email: chengwei.qiu@nus.edu.sg) or to Q.S. (email: qinghai.song@hit.edu.cn) or to S.X. (email: shumin.xiao@hit.edu.cn)

Owing to their exceptional properties in high refractive index, tunable bandgap, high efficiency of photoluminescence, and low-cost, lead halide perovskites (MAPbX$_3$, X = Cl, Br, I or their mixtures) have been very promising materials for micro- and nano-photonic devices[1–5], as well as the photovoltaics[6,7]. In past few years, a series of perovskite based nanophotonic devices are proposed and experimentally demonstrated, e.g., photodetectors[2] light-emitting diodes (LEDs)[8,9], micro and nano lasers[3,5,10–13]. The performances of perovskite solar cells, photodetectors, LEDs and lasers have been significantly improved to record high values and the corresponding practical applications are extended to displays[14–16], imaging, and sensors[17,18]. Recently, exciton has been observed at room temperature in perovskite microplate and nanorod[19–21], quickly resulting in the intense researches on the Bose-Einstein condensation and continuous-wave perovskite lasers. In addition, lead halide perovskites have also shown their great potential in nonlinear optics. The recent experiments show that the nonlinearities of lead halide perovskites are comparable to or even better than the conventional semiconductors such as GaAs[22–28]. However, despite of their exceptional nonlinear properties, lead halide perovskite based nonlinear photonic devices are still absent. This is mainly because that the lead halide perovskites are too unstable to be directly patterned with standard top-down fabrication techniques. Meanwhile, most of nonlinear processes in perovskites such as higher order harmonic generations are supposed to be consumed by their extremely strong material absorption.

Very recently, the progresses on perovskite nanophotonics bring bright hope to this field[5,15,16,29–33]. The refractive index ($n > 1.9$) of lead halide perovskite is large enough to support Mie resonances in a single nanoparticle, which can thus enhance the local electromagnetic field[5,14–16,32,33] and endow additional phase shift to incident light. Consequently, combining with the recent developments of nanofabrication techniques, the resonantly enhanced spontaneous emissions[15,16,34], light matter interaction[33], and absorption[35] in perovskite nanoparticles have been rapidly explored by different groups. By carefully designing the perovskite nanostructures and arranging them into metasurfaces, high-resolution color nanoprinting[15] and dynamic holograms[14] have also been demonstrated. Compared with their linear counterparts, the nonlinear optical processes $\left( p(E) = \varepsilon_0 (\chi^{(1)} E + \chi^{(2)} E^2 + \chi^{(3)} E^3 + \dots ) \right)$ is supposed to be more dramatically dependent on the resonant enhancement of electromagnetic field. In this sense, the incorporation of lead halide perovskite in nanostructures can fully exploit their nonlinear optical properties such as nonlinear absorption and nonlinear refractive index to construct a broad range of promising nonlinear perovskite devices[16,36].

Herein, we fabricate near-infrared MAPbBr$_3$ perovskite metasurfaces with a standard dry-etching technique and study their corresponding nonlinear processes. In contrast to the harmonic generations, we find that the three-photon luminescence can fully exploit the exceptional linear and nonlinear properties of lead halide perovskite. With the resonant enhancement of perovskite metasurface, the three-photon luminescence emission has been significantly increased by a factor of 60, enabling a series of potential applications such as nonlinear color nanoprinting and optical image encoding.

## Results

### The synthesis and characterization of perovskite films.
The lead halide perovskites films were obtained with a solution-processed spin-coating process (see "Methods")[37]. In the near-infrared spectrum, the refractive index of perovskite is relatively lower than the visible spectrum. Then the thickness plays a more essential role here to generate the near-infrared resonant modes. In our experiment, the thickness was controlled by either the concentration of precursor or the speed of spin-coating. Taking account of the grain size and surface roughness, the concentration has been optimized to 1.5 M and the thickness can be tuned from 380 to 420 nm with tiny standard deviation by controlling the speed of spin-coating (see Fig. 1a). The grain sizes and the root mean square of surface roughness, which are characterized with atomic force microscope (AFM), are less than a few hundred nanometers and 10 nm, respectively (see insets in Fig. 1a). The X-ray diffraction (XRD) spectrum in Fig. 1b shows four dominant diffraction peaks at 15.1°, 30.2°, 45° and 60°, which can be well indexed to (001), (002), (003) and (004) planes of cubic MAPbBr$_3$ perovskite. The solid and dashed lines in Fig. 1c are the photoluminescence and absorption spectra of perovskite film. Both of the bandedge and photoluminescence are consistent with the previous reports[11,17,37]. The refractive index of MAPbBr$_3$ perovskite film has also been characterized by the ellipsometer. As shown in Fig. 1d, the refractive index is well above 1.9 over a large spectral range and the absorption is negligibly small below the bandgap (2.25 eV). All of these properties show that the synthesized perovskite films have good quality and are suitable for photonic devices.

### The perovskite based nonlinear metasurfaces.
Based on the above parameters, we have numerically designed the perovskite metasurfaces (see "Methods"). Figure 2a shows its schematic picture, which consists of periodic perovskite strips. The width and lattice size are $w$ and $l$, respectively. The thickness here and below is fixed at $h = 420$ nm. Since MAPbBr$_3$ perovskites are centrosymmetric, the second harmonic generation (SHG) is neglected and the perovskite metasurfaces are designed for high order nonlinear processes[22–28]. The solid line in Fig. 2b shows the transmission spectrum of a metasurface with $w = 660$ nm and $l = 960$ nm. A resonant dip can be clearly seen at 1500 nm. The corresponding field patterns (insets in Fig. 2b) show it is a magnetic resonant mode, formed by the interference between the magnetic resonance in a single strip and the reflection of periodic structures[38,39]. As the dashed line shown in Fig. 2b, there is no strong resonances at the $3\omega$ frequency regions. As a result, the designed nonlinear processes such as THG and three-photon luminescence are mostly determined by the resonance of fundamental waves.

Compared with the single pass in a film, the resonance in perovskite nanostructure can confine light for a longer time. As a result, the incident laser is accumulated and the constructive interference shall enhance the amplitude of local electromagnetic field. Taking a THG process as an example, the amplitude is cubic proportional to the amplitude of electromagnetic field. The enhancement in the intensity of nonlinear signals (intensity) can thus be expressed as En $\propto |E_{ave}|^6 / |E_{in}|^6$. Figure 2c shows the numerically calculated enhancement factor (En) as a function of incident wavelength. The off-resonance field enhancement is negligibly small. Once the incident light is on-resonance, the local electromagnetic field is drastically increased by more than two orders of magnitude. Such resonant enhancement of electromagnetic field is quite generic in perovskite metasurface. With the variation of strip width and lattice size, a broad range of enhancement has seen in Fig. 2d, clearly indicating the robustness of designed nonlinear phenomena in perovskite metasurfaces.

Then the perovskite metasurfaces were fabricated with electron-beam lithography and an inductively coupled plasma dry etching process (see "Methods")[11]. Figure 3a shows the tilt-view scanning electron microscope (SEM) image of the perovskite

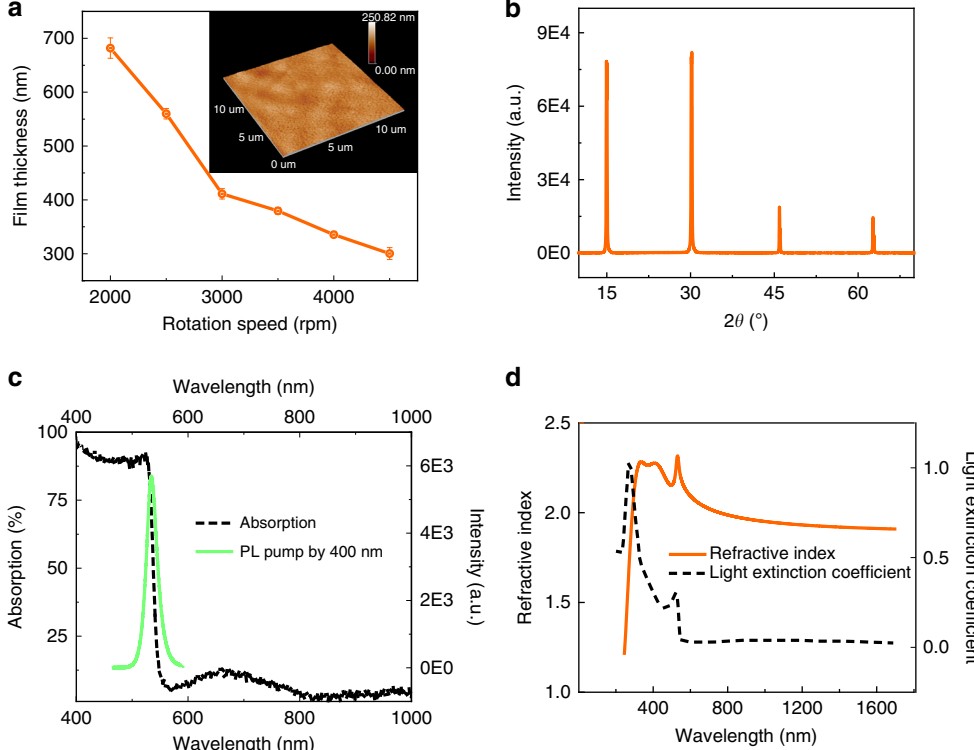

**Fig. 1** The synthesized MAPbBr$_3$ perovskite film. **a** The thickness of MAPbBr$_3$ perovskite film as a function of spin-coating speed. The inset shows the AFM image of the perovskite film. **b** The XRD spectrum of the MAPbBr$_3$ perovskite film. **c** The absorption (dashed line) and photoluminescence (solid line) curves of the perovskite film. **d** The refractive index (solid line) and light extinction coefficient (dashed line) of the synthesized perovskite film

metasurface. All the structural parameters such as lattice size ($l$), strip width ($w$), and thickness ($h$) follow the design in Fig. 2a very well. Then the transmission spectra at near infrared and visible spectrum have been experimentally recorded under a home-made microscope system by using a spectrometer coupled with two CCD cameras (see "Methods"). The infrared transmission spectrum is plotted as solid line in Fig. 3b. A resonant dip appears at 1500 nm, matching the numerical calculation and indicating the possible field enhancement well. Meanwhile, the transmission spectrum in the visible spectrum has also been recorded. Similar to the numerical simulation, there is no obvious resonances in the visible spectrum (dashed line in Fig. 3b). As a result, we can neglect the influences of Purcell effect in the visible region and double resonance, making the resonant enhancement at fundamental wavelength easy to be identified.

Then the nonlinear properties of perovskite metasurface have been characterized by exciting it with an infrared pulsed laser (100 fs pulse duration, 1000 Hz, see details in "Methods"). As schematically shown in Fig. 4a, when the metasurface is illuminated with a laser at 1500 nm in normal direction, three cyan spots and a green background can be clearly observed in the transmission or reflection directions (see the image in the inset of Fig. 4a. The emission spectrum at the cyan spot is collected and plotted as solid line in Fig. 4b. It consists of two peaks centered at 500 nm (peak-A) and 530 nm (peak-B). The intensity of peak-A is more than an order of magnitude larger than peak-B. When the detection area is outside the cyan spot, the intensity of mode-A is reduced by two orders of magnitude and the emission spectrum is now dominated by mode-B (dashed line in Fig. 4b). The open squares and open dots in Fig. 4c summarize the integrated intensity of mode-A and mode-B as a function of incident power. The fitted power slopes in log-log plot are both around 3, clearly demonstrating their nonlinear characteristics.

To identify the underlying mechanisms for the cyan spots and green background, we have further examined their dependences on the wavelength of pumping laser. All the results are shown in the inset of Fig. 4c. With the decrease of laser wavelength from 1590 to 1300 nm, we find that the position of mode-B is fixed at 530 nm, whereas the wavelength of mode-A blue-shifts linearly and always equals to 1/3 of incident laser. The polarizations of two peaks are also different. Mode-A follows the polarization of incident laser and mode-B is un-polarized. Therefore, based on the above experimental results, we can conclude that both of THG and three-photon luminescence occur in the perovskite metasurface. The first one is coherent and is diffracted by the metasurface to particular angles. The latter one is incoherent and thus uniformly distributed over a wide-angle range. Interestingly, such kind of resonant enhancement has been widely observed in a series of perovskite metasurfaces. With the decrease of lattice size $l$, the resonant dip in near infrared transmission spectrum shifts continuously from 1550 to 1440 nm (see Supplementary Note 1). The corresponding maximal enhancement factor also shifts, consistent with the above numerical simulation well.

In principle, while three-photon absorption and THG are the fifth-order process and the third-order process, respectively, they are both cubic proportional to the intensity of electromagnetic field. As mentioned above, the local field enhancement in perovskite metasurface has the ability of significantly enhancing the nonlinear signals. This is exactly what we have observed in the experiment. The off-resonance incident laser only produces weak photoluminescence. Its intensity is very close to the value of a regular perovskite film. However, once the incident laser is tuned to the resonant position (1500 nm), the intensity of three-photon luminescence is dramatically enhanced and becomes much stronger than the perovskite film. To better illustrate this effect, we defined a measure of enhancement factor as $F = I_{\text{metasurface}}/$

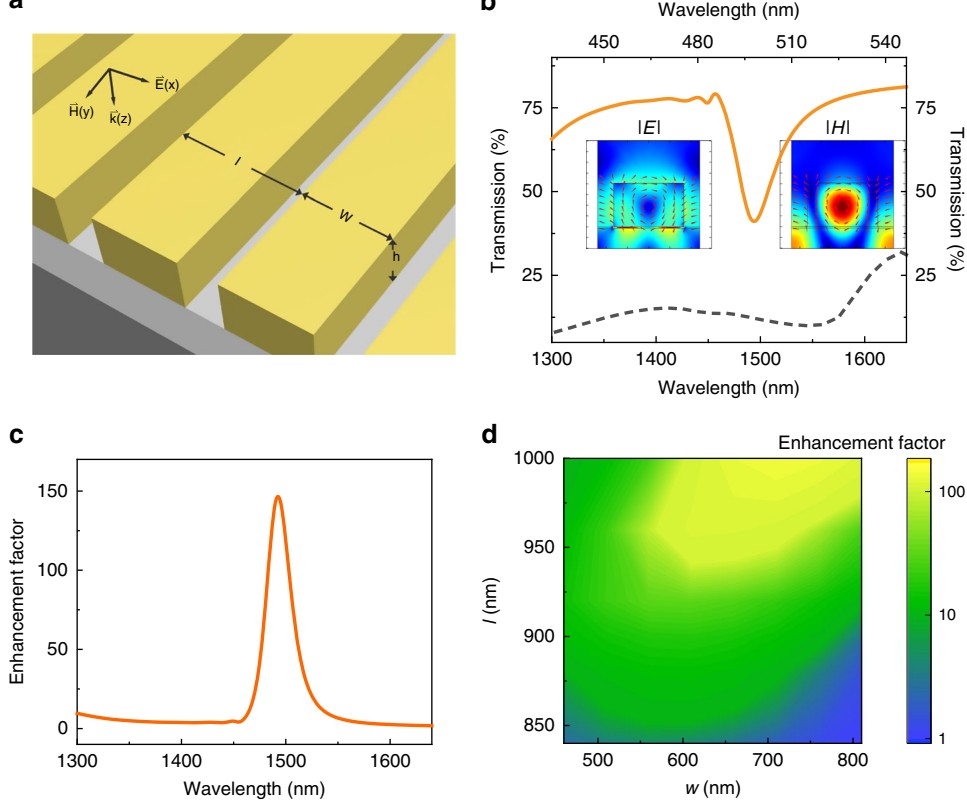

**Fig. 2** The design of MAPbBr3 perovskite metasurfaces. **a** The schematic picture of perovskite metasurface. Here the thickness of film is kept at $h = 420$ nm. The width of perovskite strip and the lattice sizes are $w$ and $l$, respectively. **b** The numerically calculated transmission spectra of perovskite metasurface with $l = 960$ nm and $w = 660$ nm at the near infrared (solid line) and visible spectra (dashed line). The insets are the electric and magnetic field distributions of resonant mode. **c** The enhancement factor of intensity in perovskite metasurface. The arrows show the directions of electric field. **d** The enhancement factor of resonant modes at a function of $l$ and $w$

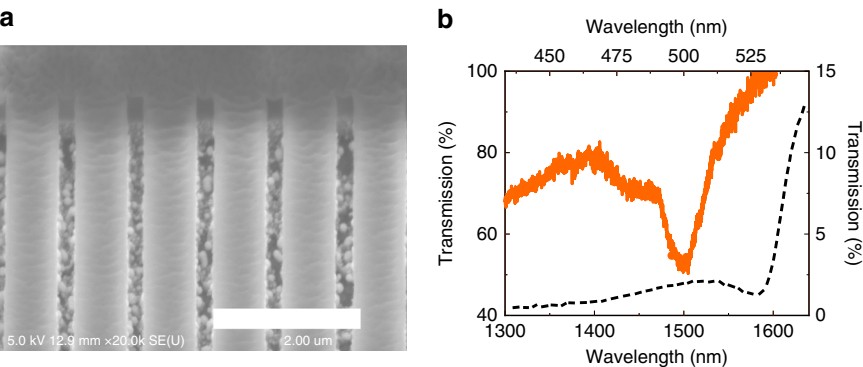

**Fig. 3** The linear properties of MAPbBr$_3$ perovskite metasurfaces. **a** The tilt-view SEM image of the perovskite metasurface, scale bar is 2 μm. **b** The experimentally measured transmission spectra at near-infrared (solid line) and visible regions (dashed line)

$I_{film}$, the normalization process of enhancement factor shows in the Supplementary Note 2. All the results are shown in Fig. 4d. The factor is more than 60 at the resonant wavelength. Such kind of enhancement is mainly attributed to the local field enhancement by taking into account the actual size and morphology of the fabricated sample. The influences of out-coupling efficiency of fluorescence[8] and Purcell factor[40] are not as strong as the refs. [8,40] (see detail discussions in the Supplementary Note 2). Similarly, the THG signals have also been enhanced at the resonant wavelength of the perovskite metasurface. However, as the open squares in Fig. 4d, the overall enhancement factor is only about 6.2. And the maximal position deviates from the resonant

wavelength. This kind of difference comes from the material absorption. When the incident laser is below 1520 nm, the corresponding THG signals are above the bandgap and are strongly absorbed by the perovskite[41]. Considering the applications of perovskite in photovoltaics, the material absorption is extremely strong and has almost fully suppressed the resonant enhancement at 1500 nm. Comparably, the three-photon luminescence is always below the bandgap and thus can fully exploit the local field enhancement. In this sense, while both of THG and three-photon luminescence can be generated and enhanced by our perovskite metasurface, the latter one can have supreme performances in practical applications.

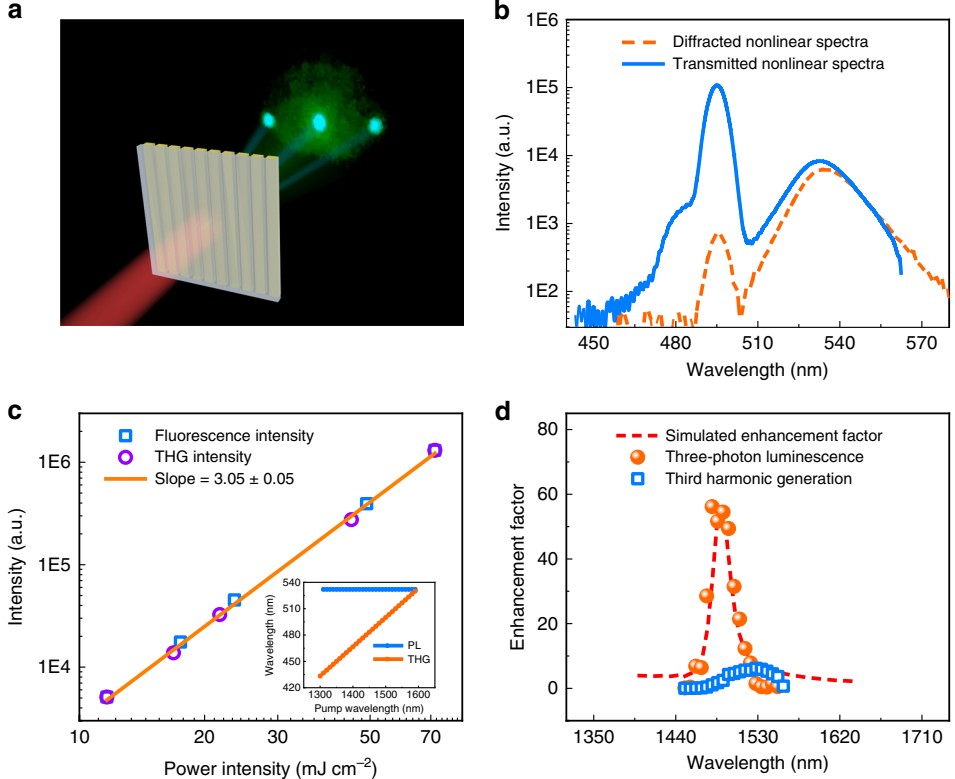

**Fig. 4** The nonlinear characterization of MAPbBr$_3$ perovskite metasurfaces. **a** The schematic picture of the nonlinear characterization. The insets are the experimentally recorded images of perovskite metasurfaces. **b** The transmitted (solid line) and diffracted (dashed line) nonlinear spectra. **c** The integrated intensity of photoluminescence as a function of incident power. Here the wavelength is fixed at 1500 nm. **d** The enhancement factor of three-photon luminescence and THG as a function of incident wavelength. Here the pumping density is kept at 30 mJ cm$^{-2}$. The dashed line is the corresponding numerically simulated resonant enhancement of incident laser by taking into account the actual size and morphology of the fabricated sample

**The nonlinear nanoprinting and optical encryption**. One of the important applications of metasurface is the color nanoprinting[42–44]. Here, the resonantly enhanced three-photon fluorescence is much stronger than the ones of un-resonant metasurface and perovskite film. As a result, the perovskite metasurfaces can also be applied as a nonlinear color nanoprinting, just like their nano-LEDs counterparts in linear region. For such kind of applications, the key parameters are the contrast and the spatial resolution. The first one relates to the enhancement factor and the latter one corresponds to the footprint of perovskite metasurface. To determine these parameters, we have reduced the period number of perovskite metasurface and studied their resonantly enhanced three-photon luminescence. All the experimental results are summarized in Fig. 5. As shown in Fig. 5a, the period of perovskite metasurfaces decreases from 40 to 20, 12, 8 and 4 in our experiment (from bottom to top). The length of perovskite strip is equal to the total width of perovskite metasurface to make a square pixel. The other structural parameters are the same as in Fig. 3.

With the decrease of period number, the collective resonance effect gets weaker and the resonant spectra are gradually dominated by the Mie scattering. As a result, the numerical calculations show that the corresponding enhancement factor reduces from round 130 to 80, 50, 30, and 18 (see dots in Fig. 5b). The solid lines in Fig. 5c are the experimentally recorded transmission spectra. The resonant wavelengths of metasurfaces with different period numbers match the numerical simulation very well. The corresponding enhancement factor are recorded and plotted in Fig. 5c. It gradually decreases from 35 to 20, 15, 7, and 3 with the reduction of period number. Compared with Fig. 5b, here the enhancement factors are influenced by the

capping layer, which is applied to protect the perovskite metasurface in experiment (see Supplementary Note 3). While the enhancement factors in Fig. 5c are not as impressive as the result in Fig. 4, as the right column shown in Fig. 5d, the resonantly enhanced photoluminescence are still good enough to be resolved from the surrounding medium. This enhanced three-photon luminescence is more straightforward in an optical image. As shown in the Supplementary Note 4, the bright green pixels can be identified from dark background (perovskite film) with the on-resonance excitation. The smallest footprint of metasurface with period of 4 is around 3.2 μm × 3.2 μm (see top-panel in Fig. 5d), giving a spatial resolution of 8000 dpi. This value is even comparable to the linear color nanoprinting systems and thus suitable for the applications including sensing, display, imaging et al.

Another important application of metasurface is the optical encryption[45–48]. While such kind of functions have been realized with conventional color nanoprinters, very complicated post-fabrication treatments are usually required, e.g., exposure to special gas and solutions[45–48]. The nonlinear perovskite metasurface has its intrinsic advantage in such applications[49]. As illustrated in Figs. 4 and 5, the intensity of three-photon luminescence from metasurfaces is only stronger than the environment when the incident laser is on resonance. Otherwise, the information will be submerged in the background. Based on this property, we have fabricated a figure with short-period perovskite metasurfaces to illustrate this potential. As the top-view SEM image shown in Fig. 5e, it is an image composed of pixels of short-period perovskite metasurfaces. With the guiding lines, we can see that encoded 'NANO' embedded in perovskite image. The high-resolution SEM images (insets in Fig. 5e) show

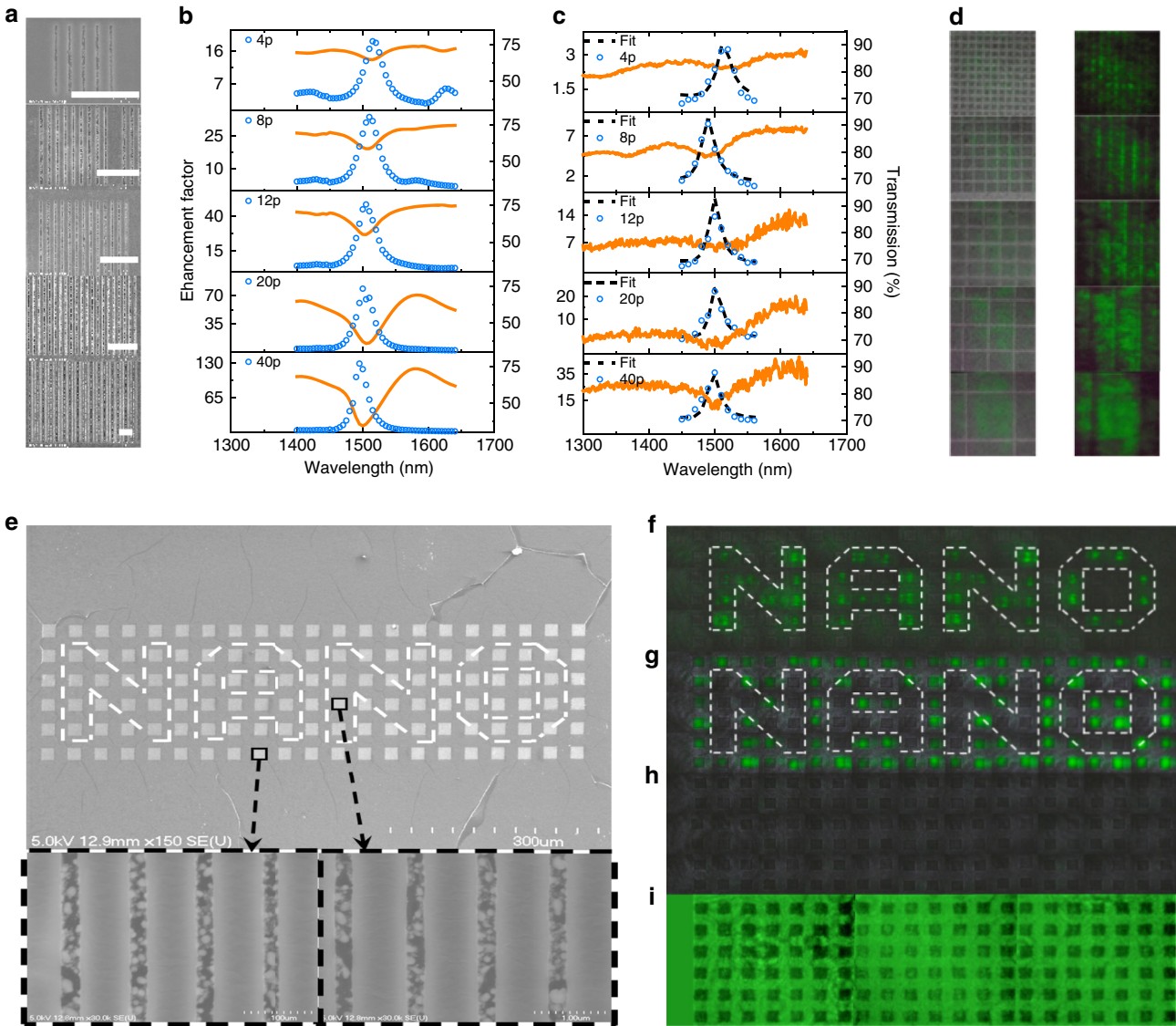

**Fig. 5** The nonlinear metasurfaces for color nanoprinting and optical encryption. **a** The top-view SEM images of perovskite metasurfaces with different period numbers, scale bar is 5 μm. **b** and **c** are the corresponding simulated and experiment transmission spectra and enhancement factors of three-photon luminescence. **d** The microscope images and the corresponding three-photon fluorescent microscope images of perovskite metasurfaces. **e** The top view SEM images of the designed metasurface for nonlinear imaging, scale bar is 300 μm. The insets are the high-resolution SEM images of the encoded information and the background, scale bar is 1 μm. Their size parameters are $l = 960$ nm and $l = 900$ nm, respectively. The width of perovskite strip is $w = l - 300$ nm. **f–i** are the corresponding nonlinear photoluminescence images under different pumping wavelengths at 1500, 1400 and 1350 nm, and linear photoluminescence image pumped by a laser at 400 nm

slight difference in lattice size (about 60 nm) between the metasurface of encoded information and background. Following the above experiments, these metasurfaces are designed for the resonances at 1500 and 1400 nm, respectively. Note that these two types of metasurfaces are so close that they are hard to be distinguished in SEM image without the guiding line (see Supplementary Fig. 4).

Figure 5f–i shows the experimental results of the perovskite metasurface under optical excitation. With the authorization, the user knows the resonant wavelength and excite the perovskite metasurface with a resonant wavelength of encoded information. Owing to the resonant enhancement, as shown in Fig. 5f, a green 'NANO' has been obtained from the dark background. The user can also get a reversed image (Fig. 5g) by resonantly enhancing the background. However, if the user is un-authorized, the perovskite metasurfaces are typically excited with off-excitation

lasers. As shown in Fig. 5h, the whole image is relatively dark and the encoded information is still un-readable when the metasurface was excited by a laser at 1350 nm (the other wavelengths are shown in Supplementary Fig. 5). With the increase of pumping power, the intensities of all pixels are increased simultaneously and the encoded information cannot be seen either. Thus we know that the encoded information can only be read with the designed wavelength. In most cases, the un-authorized attempts will excite the information by linear excitation. One example is shown in Fig. 5i with excitation at 400 nm. The displayed image is still a relatively uniform in all pixels and no information can be viewed, confirming that the encoded information cannot be cracked by single-photon excitation either. All these experimental observations are consistent with our expectation and clearly demonstrate the potential of nonlinear perovskite metasurfaces in optical encryption and optical imaging coding.

The experiment in Fig. 5 is only the proof of principle. The incident laser is designed for linear polarization with electric field perpendicular to the strip of perovskite metasurface. By exploiting the multiple design degrees of metasurfaces, more complicated incident laser can be designed, e.g., incident laser with different angular momentum[50–52]. Then the encoded information can be safer. In addition, by partially designing information, the encoded information can also be well concealed even though the unauthorized person scans the incident wavelength (see Supplementary Note 4). We must note here that the nonlinear processes within the perovskite metasurfaces are intrinsically interesting for both of fundamental researches and practical applications. In contrast to the photoluminescence, the THG signals are coherent and their local phases can be precisely controlled by the nanoantennas. While the enhancement of THG signals are limited by the material absorption, this can be simply relieved by tailoring the field distribution and the radiation of the THG signals[41]. As a result, considering the exceptional nonlinearity of the solution-processable lead halide perovskite[28], many functional devices such as nonlinear meta-lens[53], nonlinear hologram[41], and even nonlinear topological photonics[54] can also be demonstrated in a more cost-effective form.

## Discussion

In summary, we have experimentally studied the nonlinear resonances in lead halide perovskite metasurfaces. While many types of nonlinear processes can be achieved in nonlinear perovskite metasurfaces, we find that the three-photon luminescence is more efficient and experiences much stronger enhancement than the higher order harmonic generations. Based on our experimental observations, we have verified the applications of perovskite metasurfaces, the nonlinear color nanoprinting and the optical encoding. The spatial resolution can be as high as 8000 dpi, and the encoded information can only be viewed with the excitation with designed wavelength and polarizations. We believe this research shall pave an important step to the nonlinear perovskite photonics and nonlinear devices.

## Methods

**Synthesis of perovskite film**. $CH_3NH_3Br_3$ and $PbBr_2$ powders are purchased from Shanghai MaterWin New Materials co., Ltd and used as received. The $MAPbBr_3$ film is prepared by a simple spin-coating method. Specifically, 60ul of $MAPbBr_3$ precursor (1.5 M MABr and 1.5 M $PbBr_2$ dissolve in DMSO) is dropped onto the ITO layer, followed by spinning at 2900 rpm for 120 s. At the 26th second of spinning, 80 μL of chlorobenzene is quickly dropped on the film to assist in forming dense lead halide perovskite film. The prepared process is conducted in the glovebox with $Ar_2$ gas at room temperature.

**Fabrication of perovskite metasurface**. The perovskite metasurface is fabricated using an electron-beam lithography (EBL, Raith E-line) followed by an inductively coupled plasma etching (ICP, Oxford, System100 ICP180). For the lithography, a 400 nm electron-beam resist (ZEP 520A) was first spin-coated onto the perovskite film without soft-bake and then patterned by an electron beam writer with a dose 56 pC $cm^{-2}$ under an acceleration voltage of 30 kV. After developing in N50, the pattern of gratings was generated within the resist. Then the sample was etched with an Oxford Instruments Plasma Technology 380 plasma source. The chamber was pumped to reach a degree of vacuum around $10^{-9}$ Torr. Then the MAPbBr3 film was etched by chlorine gas with a 5 sccm flow rate. During the etching, $CH_4$ with a 40 sccm flow rate and $H_2$ with 10 sccm were used as a protective gas in 5mTorr pressure. The larger flow for protective gas made sure the perovskite was not damaged in the etching process. The ICP power was 150 W and the RF power was 100 W at 60 degree centigrade.

**Measurement of the 3PA-induced fluorescence**. The 3PA-induced fluorescence is measured using a custom-built microscope. A regenerative amplified femtosecond laser (Spectra-Physics, 800 nm, repetition rate 1 kHz, pulse width 100 fs, seeded by MaiTai) is coupled to an oscillating parametric amplifier (TOPAS, wavelength range: 290 nm to 2600 nm). Then the tunable laser is guided into the homemade microspore system and focused onto the sample (see Supplementary Note 6 for more detail).

**Numerical simulations**. The numerical simulations were performed with a finite element method software (COMSOL Multiphysics 4.3a). In our 2D simulation, one period of grating is calculated at frequency domain, excited by a period port with linear polarized field vertical to grating under Floquet period boundary. Meanwhile, perfect match layers (PML) are placed on the top and the bottom boundaries to minimize the reflection. The dispersion refractive index of our perovskite shows in $MAPbBr_3$ perovskite film and the refractive index of glass substrate and ZEP are fixed at 1.52 and 1.45. See Supplementary Note 7 to obtain our ITO dispersion refractive index.

## Data availability

The data that support the findings of this study are available from the authors on reasonable request, see author contributions for specific data sets.

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

## Acknowledgements

This research is financially supported by the Shenzhen Fundamental research projects (JCYJ20160427183259083), the national natural science foundation of china (NFSC) (91850204), and Shenzhen engineering laboratory on organic−inorganic perovskite devices. C.W.Q acknowledge the support from A*STAR Pharos Program (grant no. 152 70 00014, with project no. R-263-000-B91-305).

## Author contributions

S.X. and Q.S. designed the experiment. Y.W. synthesized the samples. W.S. and N.Z. fabricated the metasurfaces. Y.F. and Y.W. performed the optical characterizations. Y.F. and Y.G. did the numerical calculations. All the authors analyzed the data. Q.S., C.W.Q., and S.X. wrote the manuscript.

## Additional information

**Competing interests:** The authors declare no competing interests.

