## [Peer Review File · Nature Communications]

Reviewers' comments:

Reviewer #1 (Remarks to the Author):

The manuscript "Resonance-Enhanced Upconversion in Structured Lead Halide Perovskites" by Yunbin Fan et al. reports on the enhancement of nonlinearly excited photoluminescence and third-harmonic generation from halide perovskite (MAPbBr₃) metasurface. Generally, the topic of nanophotonics with halide perovskites is of high interest owing to the fascinating optical properties of this family of materials, which can be further boosted by novel photonic designs. In this work, the authors timely showed how a perovskite metasurface can enhance the third-order nonlinear response of the perovskite. Moreover, they show the direct application of this effect in optical encryption. As a result, I believe this manuscript can be published in Nature Communications after addressing my comments below:

- The authors wrote "Herein, we fabricate near-infrared MAPbBr₃ perovskite metasurfaces with a standard dry-etching technique and study their corresponding nonlinear processes for the first time." However, the nonlinearly excited three-photon PL in perovskite metasurface was investigated in Ref[16] as well. So, the authors should provide a more detailed comparison with this work.
- Instead, I recommend to stress the novelty of the enhanced third-harmonics generation from the perovskite metasurface.
- The authors explained lower values for THG enhancement by absorption in perovskite. However, the enhancement should be governed by resonances at a fundamental wavelength. I would expect that some additional effects (outcoupling [Nature, 562(7726), p.249 (2018)] or Purcell effect [Nano Letters 18 (2), 1185-1190 (2018)]) at the emission wavelengths should contribute to PL and THG.
- The term "Upconversion" is widely used for anti-stokes luminescence excited via real states (see e.g. [Wang, F. and Liu, X., 2009. Recent advances in the chemistry of lanthanide-doped upconversion nanocrystals. Chemical Society Reviews, 38(4), pp.976-989] or [Kakavelakis, G., Petridis, K. and Kymakis, E., 2017. Recent advances in plasmonic metal and rare-earth-element upconversion nanoparticle doped perovskite solar cells. Journal of Materials Chemistry A, 5(41), pp.21604-21624.]), rather than for nonlinear multiphoton absorption via virtual states. I suggest to check whether it is possible to use this term in the manuscript context.
- A bracket is missed here: "(see the image in the inset of Fig. 4a".

Reviewer #2 (Remarks to the Author):

The paper by Fan et al. reports on the possibility to employ metasurfaces to enhance the nonlinear optical properties of MAPbX₃ perovskites. I think their outcomes are quite interesting, especially for their applications in the field of optical encryption. The results are explained in a convincing manner and I recommend the publication of this manuscript after minor revision.

There are a few changes that should, in my opinion, be made to significantly improve the manuscript:

- The title of the manuscript is misleading. It is not clear that the resonances claimed are due to the presence of metasurfaces, which should be stressed more.
- Did the authors measure the eventual presence of an enhancement at the excitation wavelength of 1500nm for the un-structured (without metasurfaces) perovskite material? If so, the enhancement factors should be normalized to that values.
- Resonances due to the excitonic nature of perovskites nanocrystals have been reported by Manzi et al. in Nature Communications 9 (1), 1518. Although in this manuscript the excitation sources do not have enough energy to generate multiple excitons, the authors should consider the possibility of an enhancement (if present also without the metasurfaces) in line with what reported previously.

Reviewer #3 (Remarks to the Author):

The manuscript by Fan et al. reports on the resonance selectivity in three-photon-induced photoluminescence (PL) from an engineered metasurface based on a hybrid halide perovskite, MAPbBr₃. While the work is straightforward and rather interesting in terms of optical encoding/decoding, it should be published in a more specialized journal because of the lack of general interest. Indeed, it does not have any surprising result to warrant the publication in any of Nature's sister journals.

There is one minor mistake: In page 5, line 149, "In principle, both of 3PA and THG are cubic proportional to the amplitude of electromagnetic field." They are cubic in the "intensity" of the field, not the "amplitude".

The reviewer also suggests that the authors should describe the fundamental difference between 3PA and THG, when they resubmit the revision elsewhere: One is the fifth-order process, whereas the other is the third-order process.

They should also describe why the experimental results in Fig. 5(c) is seriously different from the simulation in Fig. 5(b). Although the current study is a concept study, the real enhancement factor is not impressive at the present level

Reply to Reviewer 1:

We really appreciate the reviewer for the careful review and valuable comments. We also thank the reviewer for the suggestion of acceptance. Following the reviewer's comments, we have carefully revised our manuscript. All the replies have been added in the manuscript accordingly.

Comment-1. The authors wrote "Herein, we fabricate near-infrared MAPbBr₃ perovskite metasurfaces with a standard dry-etching technique and study their corresponding nonlinear processes for the first time." However, the nonlinearly excited three-photon PL in perovskite metasurface was investigated in Ref [16] as well. So, the authors should provide a more detailed comparison with this work.

Our Reply: We really appreciate the reviewer for pointing out this valuable reference. We agree with the reviewer that Ref. [16] has also discussed the three-photon PL in perovskite metasurface. Despite similar three-photon responses have been observed in Ref. [16] and this work, their fundamental bases and possible applications are quite different. In Ref. [16], the wavelength of pumping laser is 1050 nm and the bandedge of material is around 770 nm. The three-photon process is realized by the intrinsic material properties of perovskites, i.e. the resonance in photobleaching at 3.2 eV. The resonance of perovskite metasurface is designed for the modification of local density of states (LDOS) around the PL wavelength. In our research, the three-photon PL is generated by the field enhancement of the incident laser, which is generated by the resonant mode of perovskite metasurface. Compared with the intrinsic material properties, the nanostructure induced enhancement is dependent on the structural parameters and thus can be tuned to any designed wavelength, e.g. 1500 nm-1600 nm. This is the fundamental basis for the applications of this research. With the resonance enhancement of the pumping laser, we have demonstrated the applications of high resolution (8000 dpi) nonlinear display and the optical encryption. The encoded information "NANO" can only be viewed when the incident laser is on resonance.

The most valuable part of this comment is that Reviewer-1 has provided us a new research direction. Since both of the resonances of nanostructures (for incident laser) and the materials (for three-photon) can greatly enhance the three-photon PL process, the enhancement factor shall be more dramatic if these two effects are combined together in the perovskite metasurface. The importance of Ref. 16 has been highlighted in para-2, page-7 of the revised manuscript. The corresponding experiment will be done in near future. "We must note here that the nonlinear processes within the perovskite metasurfaces are intrinsically interesting for both of fundamental researches and practical applications. As pointed out by Ref. [16], the intrinsic material properties such as the state in photobleaching region (3.2 eV) can also enhance the three-photon luminescence by tens of times. Similarly, the material enhancement can also be realized with the multi-exciton resonances [52]. Consequently, the combination of

structural enhancement and material enhancement in perovskite metasurface can further improve the nonlinear responses.”

To avoid the unnecessary confusion, we have also removed the statement of “for the first time” in the revised manuscript.

Comment-2. I recommend to stress the novelty of the enhanced third-harmonics generation from the perovskite metasurface.

Our Reply: We thank the reviewer for this constructive and deep-insight recommendation. We fully agree with the reviewer that the THG process in perovskite metasurface is also attractive and worth to emphasize. The perovskite materials can provide the exceptional nonlinear properties and cost-effective fabrication process, whereas the metasurface is able to precisely control the wavefront of the THG waves. As a result, a large number of nonlinear devices can be realized in a higher efficiency and more cost-effective form. This will be essential for practical applications. Following the reviewer’s suggestion, we have also stressed the novelty of the enhanced THG process from perovskite metasurface in para-2, page-7. “In addition, the enhanced THG processes in perovskite metasurface is also interesting. In contrast to the photoluminescence, the THG signals are coherent and their local phases can be precisely controlled by the nanoantennas. While the enhancement of THG signals are limited by the material absorption, this can be simply relieved by tailoring the field distribution and the radiation of the THG signals [40]. As a result, considering the exceptional nonlinearity of the solution-processable lead halide perovskite [28], many functional devices such as nonlinear meta-lens [53], nonlinear hologram [40], and even nonlinear topological photonics [54] can also be demonstrated in a more cost-effective form.”

Comment-3 The authors explained lower values for THG enhancement by absorption in perovskite. However, the enhancement should be governed by resonances at a fundamental wavelength. I would expect that some additional effects (outcoupling [Nature, 562(7726), p.249 (2018)] or Purcell effect [Nano Letters 18 (2), 1185-1190 (2018)]) at the emission wavelengths should contribute to PL and THG.

Our Reply: We thank the reviewer for this valuable comment and the suggestion of two references. These two references have been added as Ref. [8] and Ref. [39] in the revised manuscript.

The reviewer is absolutely correct that the out-coupling efficiency and the Purcell factors in Refs. 8 and 39 are possible to play an essential role in the spontaneous emission. Following the reviewer’s suggestion, we have numerically calculated the outcoupling efficiency of our metasurface with the same process as Ref. [8]. Basically, six point dipoles are embedded in the metasurfaces (see Fig. R1(a)). The corresponding far field patterns are recorded and compared with the same dipoles in perovskite film. All the numerical calculation results are

summarized in Fig. R1(b)-Fig. R1(d). It is easy to see that the enhancement factors are around 1 in all directions. This shows that the influence of output coupling efficiency on the enhancement factor is much smaller than the enhancement via the resonance at the fundamental wavelength.

Figure R1. The influence of output coupling efficiency of spontaneous emission. (a) The schematic picture and the locations of six dipoles in perovskite metasurface. (b) – (d) The enhancement factor for the output coupling efficiency of dipoles align in x, y, and z directions.

The Purcell factor around the photoluminescence wavelength region has also been numerically studied. As shown in Fig. R2, the transmission spectrum at the visible spectrum indeed has a dip around the spontaneous emission wavelength range. However, as the perovskite metasurface is designed for the near infrared wavelength, the higher order resonances are quite weak and can only be barely seen from the background (1-2% difference). As a result, its influence on the spontaneous emission cannot be as significant as Ref. [39].

Figure R2. The experimental calculated transmission spectra in the near infrared (solid line) and visible spectrum (dashed line).

Our experimental results are also consistent with the above numerical simulation. As shown in Fig. 4(b) in the main manuscript, there is no obvious additional enhancement around 525 nm even though the spectra are plotted in log scale. Therefore, we can confirm that the output coupling efficiency and the Purcell factor are not as significant as the resonant enhancement of the incident laser.

In the revised manuscript, we have added the above discussion in para-1, page-2. “Such kind of enhancement is mainly attributed to the local field enhancement by taking into account the actual size and morphology of the fabricated sample. The influences of out-coupling efficiency of fluorescence [8] and Purcell factor [39] are not as strong as the Refs. [8, 39] (see detail discussions in the Supplementary Information Note 2).”. The corresponding numerical results and detail discussions are added into the Supplementary Information Note 2.

Comment-4 The term “Upconversion” is widely used for anti-stokes luminescence excited via real states (see e.g. [Wang, F. and Liu, X., 2009. Recent advances in the chemistry of lanthanide-doped upconversion nanocrystals. *Chemical Society Reviews*, 38(4), pp.976-989] or [Kakavelakis, G., Petridis, K. and Kymakis, E., 2017. Recent advances in plasmonic metal and rare-earth-element upconversion nanoparticle doped perovskite solar cells. *Journal of Materials Chemistry A*, 5(41), pp.21604-21624.]), rather than for nonlinear multiphoton absorption via virtual states. I suggest to check whether it is possible to use this term in the manuscript context.

Our Reply: We thank the reviewer for this valuable suggestion. We agree with the reviewer that the upconversion is not very accurate and will lead to unnecessary confusion. To make the statement accurately and avoid confusion, we have replaced all the “upconversion” and “three-photon upconversion” into “three-photon luminescence”. The latter one is more accurate to describe the nonlinear multiphoton absorption via virtual states.

Reply to Reviewer 2:

We thank the reviewer for the very careful review and the valuable suggestions for revision. We also thank the reviewer for the recommendation for publishing on Nature Communications. Based on his/her suggestion, we have carefully revised the manuscript and all the comments have been addressed accordingly.

Comment-1. The title of the manuscript is misleading. It is not clear that the resonances claimed are due to the presence of metasurfaces, which should be stressed more.

Our Reply: We thank the reviewer for this valuable comment. In our research, the enhancement is caused by the resonant mode of the perovskite metasurface at the wavelength of incident laser. This can be seen from the dependence of maximal enhancement factor on the lattice sizes in Fig. R3 below. With the change of the nanostructures, the resonant wavelength and the enhancement gradually shift from ~ 1320 nm to 1590 nm. This is different from the enhancement of material resonance, which is usually fixed at particular wavelength.

We think that the “upconversion” in title is indeed confusing. It usually leads the general readers to anti-stokes luminescence excited via real states such as the rare-earth-element upconversion nanoparticles. To avoid the possible confusion, we have replaced the “upconversion” in the title and the whole manuscript with “three-photon luminescence”. The title of our manuscript is also changed to “Resonance-enhanced Three-photon Luminescence via Lead Halide Perovskite Metasurfaces”. The results in Fig. R3 has also been added in Supplementary Information Note 1.

Figure R3: The transmission (a) and three-photon luminescence enhancement factor (b) of MAPbBr₃ perovskite metasurfaces. The lattice size of the metasurface changes from $l = 840$ nm, $l = 880$ nm, $l = 900$ nm, $l = 920$ nm, $l = 960$ nm, to $l = 1000$ nm. Here the width w is $w = l - 300$ nm. With the change of lattice size, the resonant wavelength shifts to shorter wavelength. The maximal enhancement also shifts simultaneously and always match the

resonant position.

Comment-2. Did the authors measure the eventual presence of an enhancement at the excitation wavelength of 1500nm for the un-structured (without metasurfaces) perovskite material? If so, the enhancement factors should be normalized to that values.

Our Reply: We thanks the reviewer for this very careful review. The reviewer is absolutely right that the enhancement factor should be normalized to the emission from unstructured perovskite films. In our manuscript, we indeed calculated the enhancement factor by measuring the photoluminescence intensities from both of metasurface and un-structured perovskite film. Taking the three-photon luminescence as an example, Fig. 4(d) shows the experimentally recorded enhancement factor. To get these values, we kept the pumping density and measured the emissions from metasurface (orange line in Fig. R4) and the perovskite film (blue line in Fig. R4). The dots in Fig. 4(d) in the main manuscript is obtained by normalizing the orange line with the blue line. The THG signals are also obtained with a similar process.

To avoid the possible confusion, we have added the results in Fig. R4 into the Supplementary Information Note 2 to clarify the normalization process.

Figure R4. The integrated photoluminescence intensity from the perovskite metasurface (orange line) and the perovskite film (blue line). Figure 4(d) is obtained by normalizing the emission from metasurface with the blue line.

Comment-3. Resonances due to the excitonic nature of perovskites nanocrystals have been reported by Manzi et al. in Nature Communications 9 (1), 1518. Although in this manuscript the excitation sources do not have enough energy to generate multiple excitons, the authors should consider the possibility of an enhancement (if present also without the metasurfaces) in line with what reported previously.

Our Reply: We thank the Reviewer for the suggestion of this very important reference. We have added this reference as Ref. [52] in the revised manuscript. The multiple exciton has the ability of enhancing the two-photon photoluminescence intensity by orders of magnitude. We

agree with the reviewer that it is very important to consider this possibility. Following the reviewer's suggestion, we have also experimentally verified the wavelength dependence of two-photon luminescence from our perovskite film. All the results are shown in Fig. R5 below. With the decrease of pumping wavelength, there is a dramatic increase at around 720 nm. This is also caused by the excitonic resonance at ~ 3.4 eV, which can greatly enhance the multi-photon absorption process. Such kind of additional enhancement is quite interesting and worth to study in future. For our current research, as the enhancement factor is calculated by normalizing with the perovskite film, the material enhancement won't change the final experimental results.

Since both the pure material resonance from multiple exciton resonance and the structural resonance can enhance the nonlinear processes, the combination of two effects (designing the structural resonance for the excitation of multi-exciton resonance) can further improve the nonlinear signals by orders of magnitude. This is very important and we will focus on such studies in near future. In the revised manuscript, we have added the corresponding discussion in para-2, page-7. “Similarly, the material enhancement can also be realized with the multi-exciton resonances [52]. Consequently, the combination of structural enhancement and material enhancement in perovskite metasurface can further improve the nonlinear responses.” The results in Fig. R5 has also been added into the Supplementary Information Note 5.

Figure R5: The wavelength dependent two-photon luminescence. Due to the presence of an excitonic resonance at around 3.4 eV, the two-photon absorption is also significantly enhanced.

Reply to Reviewer 3:

We thank the reviewer for the careful review and valuable suggestion. In the revised manuscript, we have fully addressed all the comments accordingly. Based on his/her suggestions, the quality of this research has been significantly improved.

The main comment raised by reviewer-3 is the general interest of this work. We thank the reviewer for this deep insight comment. As the reviewer knows, lead halide perovskites have been incorporated with extraordinary success in photovoltaics. Such successes come from the fact that lead halide perovskite can provide similar or better power conversion efficiency (PCE) while significantly reduce the device cost. Soon after, with the improvements in quantum efficiency and light coupling efficiency, similar successes have also been achieved in lead halide perovskite based light emitting diodes (LEDs) and photodetectors. Now, the recent researches reveal that the nonlinearity of lead halide perovskite is comparable to or even superior than Si and III-V direct bandgap semiconductors again. In this context, it has been recognized that it is really the turn of perovskite nonlinear photonics.

With this background, we started to explore the lead halide perovskite based nonlinear photonic devices. In principle, the performances of nonlinear devices are determined by two factors, i.e. the nonlinearity of materials and the geometrical distribution of the nonlinear materials. While the nonlinear processes have been intensively studied in perovskite, most of these researches are limited in material side even though the signals can be greatly enhanced by the material resonances (such as Ref. 16 and Ref. 52). In our research, we are working on the influences of geometrical distributions of materials on the nonlinear process. Compared with the material enhancement, the resonances of perovskite metasurface is an additional degree to further enhance the nonlinear signals. Most importantly, it is more flexible than the material resonance (fixed at particular wavelength) and can be precisely tailored via the nanostructures. Therefore, we have experimentally demonstrated not only the greatly enhanced three-photon luminescence but also the practical applications such as nonlinear light emitting devices, high resolution nonlinear display, and the optical encryption.

From the aspect of material science, this research shall pave an important step to the applications in nonlinear perovskite photonics. Considering the exceptional nonlinearity of the solution-processable lead halide perovskite and the possibility of reducing the material absorption of THG, this research can also lead to the developments of many functional devices such as nonlinear meta-lens, nonlinear hologram, and even nonlinear topological photonics in a more cost-effective form. Therefore, as pointed out in a recent perspective paper (Ref. 28), now it should be the time to study and develop perovskite based nonlinear photonic devices.

Comment-1. There is one minor mistake: In page 5, line 149, “In principle, both of 3PA and

THG are cubic proportional to the amplitude of electromagnetic field.” They are cubic in the “intensity” of the field, not the “amplitude”.

Our Reply: Thank the reviewer for this careful review. The reviewer is correct that 3PA and THG are cubic proportional the intensity instead of amplitude of the incident laser. In the revised manuscript, we have changed the sentence in para-1, page-5. “In principle, while three-photon absorption and THG are the fifth-order process and the third-order process, respectively, they are both cubic proportional to the intensity of electromagnetic field.”

Comment-2. The reviewer also suggests that the authors should describe the fundamental difference between 3PA and THG, when they resubmit the revision elsewhere: One is the fifth-order process, whereas the other is the third-order process.

Our Reply: We thanks the reviewer for this fundamental suggestion. The reviewer is correct that three-photon absorption is the fifth-order process, whereas the THG is the third order process. This can be directly seen from the equations.

For an isotropic medium in vacuum illuminated by fundamental electronic field amplitude $E(\omega)$ with the same frequency ω , complex amplitudes of the nonlinear polarization can be written as

$$P^{(3)}(3\omega) = \varepsilon_0 \chi^{(3)} E^3(\omega).$$

In case of three-photon absorption, the change of irradiance with depth into the sample as below

$$\frac{dI(z)}{dz} = -\frac{5\omega}{2n_0^3 c^3 \varepsilon_0^3} \chi^{(5)} I^3(z).$$

Here n_0 is the refractive index of an isotropic medium at the irradiance wavelength, c is the speed of light and ε_0 is the permittivity of vacuum. Following the reviewer’s suggestion, we have added the information in para-1, page-5 in the revised manuscript. “In principle, while three-photon absorption and THG are the fifth-order process and the third-order process, respectively, they are both cubic proportional to the intensity of electromagnetic field.”

Comment-3. They should also describe why the experimental results in Fig. 5(c) is seriously different from the simulation in Fig. 5(b). Although the current study is a concept study, the real enhancement factor is not impressive at the present level

Our Reply: We thanks the reviewer for this careful review and valuable suggestion. Based on the reviewer’s comment, we have carefully checked our experimental results and the numerical simulation. The difference comes from the capping layer on top of our sample. In our simulation, we directly simulated the perovskite nanostructures and got the enhancement factor. In experiment, considering that the perovskite is not stable, we have to use a cap layer (ZEP) to protect the nanostructures in room temperature and ambient environment. Compared with the air, the capping layer reduce the difference of refractive indices of perovskite

nanostructure and the surrounding medium. The reduced refractive index difference changes the local field enhancement and thus reduces the enhancement factor. If the capping layer is considered, as the dashed lines shown in Figure R6c below, the experimental results match the numerical simulation very well if the capping layer is considered. In the revised manuscript, we have replaced the data in Fig. 5 and added the corresponding discussion in para-1, page-6. “Compared with Fig. 5b, here the enhancement factors are influenced by the capping layer, which is applied to protect the perovskite metasurface in experiment (see Supplementary Information Note 3).” The detail discussions have been added into the Supplementary Information Note 3.

We agree with the reviewer that the enhancement factor in Fig. 5 is not that impressive. This is mainly caused the reduction of periodicity of metasurface. For metasurface with infinite periodicity number, the enhancement factor can be as large as 150 as shown in Fig. 2(c). However, in order to achieve a large number of practical applications such as optical encryption, the spatial resolution should be taken into account. By decreasing the periodicity number, the enhancement factor will reduce progressively. The smallest one is realized with only four strips ($N = 4$). But it is still clear enough to be distinguished from the perovskite film (background, see top row in Fig. R6d below) and meet the needs of practical applications. The intensity of nonlinear signals can also be significantly enhanced if the other material properties are considered. For example, the multi-exciton resonance or state in photobleaching region can further improve the enhancement of three-photon luminescence. We have also discussed this possibility in para-2, page-7 of the revised manuscript. “As pointed out by Ref. [16], the intrinsic material properties such as the state in photobleaching region (3.2 eV) can also enhance the three-photon luminescence by tens of times. Similarly, the material enhancement can also be realized with the multi-exciton resonances [52]. Consequently, the combination of structural enhancement and material enhancement in perovskite metasurface can further improve the nonlinear responses.”

Figure R6 | The nonlinear metasurfaces for color nanoprinting and optical encryption. (a)

The top-view SEM images of perovskite metasurfaces with different period numbers, scale bar is $5\ \mu\text{m}$. **(b)** and **(c)** are the corresponding simulated and experiment transmission spectra and enhancement factors of three-photon luminescence. **(d)** The microscope images and the corresponding three-photon fluorescent microscope images of perovskite metasurfaces. **(e)** The top view SEM images of the designed metasurface for nonlinear imaging, scale bar is $300\ \mu\text{m}$. The insets are the high-resolution SEM images of the encoded information and the background, scale bar is $1\ \mu\text{m}$. Their size parameters are $l = 960\ \text{nm}$ and $l = 900\ \text{nm}$, respectively. The width of perovskite strip is $w = l - 300\ \text{nm}$. **(f) – (i)** are the corresponding nonlinear photoluminescence images under different pumping wavelengths at $1500\ \text{nm}$, $1400\ \text{nm}$ and $1350\ \text{nm}$, and linear photoluminescence image pumped by a laser at $400\ \text{nm}$.

REVIEWERS' COMMENTS:

Reviewer #1 (Remarks to the Author):

The authors have carefully revised the most of critical parts of the manuscript and improved its overall quality. The results sound technically and provide full understanding of the processes behind the observed phenomena. Generally, I satisfied with all the answers except the first one, because there is still a serious problem related to the comparison with previous studies.

a) The authors incorrectly wrote that "As pointed in Ref.[16], the intrinsic material properties such as the state in photobleaching region (3.2 eV) can enhance the three-photon luminescence by tens of times.". However, even the title of this paper "Multifold emission enhancement in nanoimprinted hybrid perovskite metasurfaces" clearly implies that the three-photon luminescence enhancement originates from optical resonances in a perovskite metasurface at wavelength 1050 nm rather than from any material effects. Moreover, in this paper, exactly the same enhancement factor (up to 70 times) for three-photon luminescence was observed owing to resonant modes excitation. Thus, the discussion on page 7 should be revised, or put somewhere before the results on optical encoding.

b) In this regard, the modified title of the submitted manuscript ideologically resembles that in Ref.[16]. However, the manuscript contains very important and novel results on optical encoding. I suggest to stress this in title directly to avoid considerable overlapping with previous papers. For example, "Resonance-enhanced Three-photon Luminescence via Lead Halide Perovskite Metasurfaces for Optical Encoding", or something like that.

I believe that the paper deserves publication in Nature Communication after the addressing the above-mentioned comments.

Reviewer #2 (Remarks to the Author):

The authors have adequately replied to my original comments and provided satisfactory answers to the points I have raised. I suggest acceptance of this work in its current form.

Reply to Reviewer 1:

We really appreciate the reviewer for the careful review and valuable comments. We also thank the reviewer for the high evaluation. Following the reviewer's comments, we have carefully revised our manuscript and title. All the replies have been added in the manuscript accordingly.

Comment-1. The authors incorrectly wrote that “As pointed in Ref. [16], the intrinsic material properties such as the state in photobleaching region (3.2 eV) can enhance the three-photon luminescence by tens of times.”. However, even the title of this paper “Multifold emission enhancement in nanoimprinted hybrid perovskite metasurfaces” clearly implies that the three-photon luminescence enhancement originates from optical resonances in a perovskite metasurface at wavelength 1050 nm rather than from any material effects. Moreover, in this paper, exactly the same enhancement factor (up to 70 times) for three-photon luminescence was observed owing to resonant modes excitation. Thus, the discussion on page 7 should be revised, or put somewhere before the results on optical encoding.

Our Reply: We really appreciate the reviewer for this suggestion. We delete the discussion about the properties effect and revise the manuscript in paragraph 2, page 7. “The experiment in Fig. 5 is only the proof of principle. The incident laser is designed for linear polarization with electric field perpendicular to the strip of perovskite metasurface. By exploiting the multiple design degrees of metasurfaces, more complicated incident laser can be designed, e.g. incident laser with different angular momentum [50-52]. Then the encoded information can be safer. In addition, by partially designing information, the encoded information can also be well concealed even though the unauthorized person scans the incident wavelength (see Supplementary Note 4). We must note here that the nonlinear processes within the perovskite metasurfaces are intrinsically interesting for both of fundamental researches and practical applications. In contrast to the photoluminescence, the THG signals are coherent and their local phases can be precisely controlled by the nanoantennas. While the enhancement of THG signals are limited by the material absorption, this can be simply relieved by tailoring the field distribution and the radiation of the THG signals [40]. As a result, considering the exceptional nonlinearity of the solution-processable lead halide perovskite [28], many functional devices such as nonlinear meta-lens [53], nonlinear hologram [40], and even nonlinear topological photonics [54] can also be demonstrated in a more cost-effective form.”

Comment-2. In this regard, the modified title of the submitted manuscript ideologically resembles that in Ref. [16]. However, the manuscript contains very important and novel results on optical encoding. I suggest to stress this in title directly to avoid considerable overlapping with previous papers. For example, “Resonance-enhanced Three-photon Luminescence via Lead Halide Perovskite Metasurfaces for Optical Encoding”, or something like

that.

Our Reply: We really thank the reviewer for the generous suggestion. Based on this valuable suggestion, we change the title to “Resonance-enhanced Three-photon Luminescence via Lead Halide Perovskite Metasurfaces for Optical Encoding”.